# Universal holonomic quantum gates over geometric spin qubits with polarised microwaves

Kodai Nagata[1], Kouyou Kuramitani[1], Yuhei Sekiguchi [1] & Hideo Kosaka[1]

A microwave shares a nonintuitive phase called the geometric phase with an interacting electron spin after an elastic scattering. The geometric phase, generally discarded as a global phase, allows universal holonomic gating of an ideal logical qubit, which we call a geometric spin qubit, defined in the degenerate subspace of the triplet spin qutrit. We here experimentally demonstrate nonadiabatic and non-abelian holonomic quantum gates over the geometric spin qubit on an electron or nitrogen nucleus. We manipulate purely the geometric phase with a polarised microwave in a nitrogen-vacancy centre in diamond under a zero-magnetic field at room temperature. We also demonstrate a two-qubit holonomic gate to show universality by manipulating the electron—nucleus entanglement. The universal holonomic gates enable fast and fault-tolerant manipulation for realising quantum repeaters interfacing between universal quantum computers and secure communication networks.

[1] Yokohama National University, 79-5 Tokiwadai, Hodogaya, Yokohama 240-8501, Japan. These authors contributed equally: Kodai Nagata, Kouyou Kuramitani.  Correspondence and requests for materials should be addressed to H.K. (email: kosaka-hideo-yp@ynu.ac.jp)

The geometric phase[1–24], which is closely related to the topological phase, is currently a central issue not only in quantum physics but also in various scientific fields including optics, electronics, nanotechnology, and materials science. In quantum physics, the holonomic quantum gate manipulating purely the geometric phase in the degenerate ground state system is believed to be an ideal way to build a fault-tolerant universal quantum computer. The concept of the geometric phase was first proposed by Pancharatnam[1] for light polarisation, which is a typical degenerate ground state system with inner degree of freedom. Since then, an adiabatic geometric phase was formalised by Berry[2] and then was generalised into a non-abelian geometric phase by Wilczek and Zee[3] as well as into nonadiabatic geometric phases by Anandan[4]. Meanwhile, their application to fault-tolerant holonomic quantum computing[5–7,9,10] was proposed, and the geometric phase gate or holonomic quantum gate has been experimentally demonstrated in a superconducting qubit[12,13], in trapped ions[14,15] in a quantum dot[16], in molecular ensembles[17,18], and recently in a nitrogen-vacancy (NV) centre in diamond[19–24].

However, these experiments required microwaves[19,20] or light waves[21,24] in two frequencies differing by the energy gap for manipulating the nondegenerate subspace, leading to the degradation of gate fidelity due to the unwanted interference of the dynamic phase. To avoid such interference, we used a degenerate subspace of the triplet spin qutrit in spin-1 system to form a logical qubit called a geometric spin qubit[11,22,23], where the geometric phase was manipulated with a light resonant to the optical excited state as an ancillary level[22–24]. This method allowed fast and precise geometric gates, but could only control a single qubit made of an electron spin at a temperature below 10 K and the gate fidelity was limited by radiative relaxation.

The advantage of a polarised microwave[25,26] for the manipulation of an electron spin has been demonstrated with the use of a circularly polarised microwave generated by two crossed wires (Fig. 1a, b) to overcome the limit of the rotation-wave approximation[26].

We here demonstrate nonadiabatic and non-abelian holonomic quantum gates over the geometric spin qubit on an electron or nitrogen nucleus with a polarised microwave in a nitrogen-vacancy centre in diamond under a zero-magnetic field at room temperature. We also demonstrate a two-qubit holonomic gate to show universality by manipulating the electron−nucleus entanglement.

## Results

**Principle of geometric spin manipulation by polarised microwaves.** We start by showing the analogy between microwave polarisation and geometric spin polarisation (Fig. 1c). The

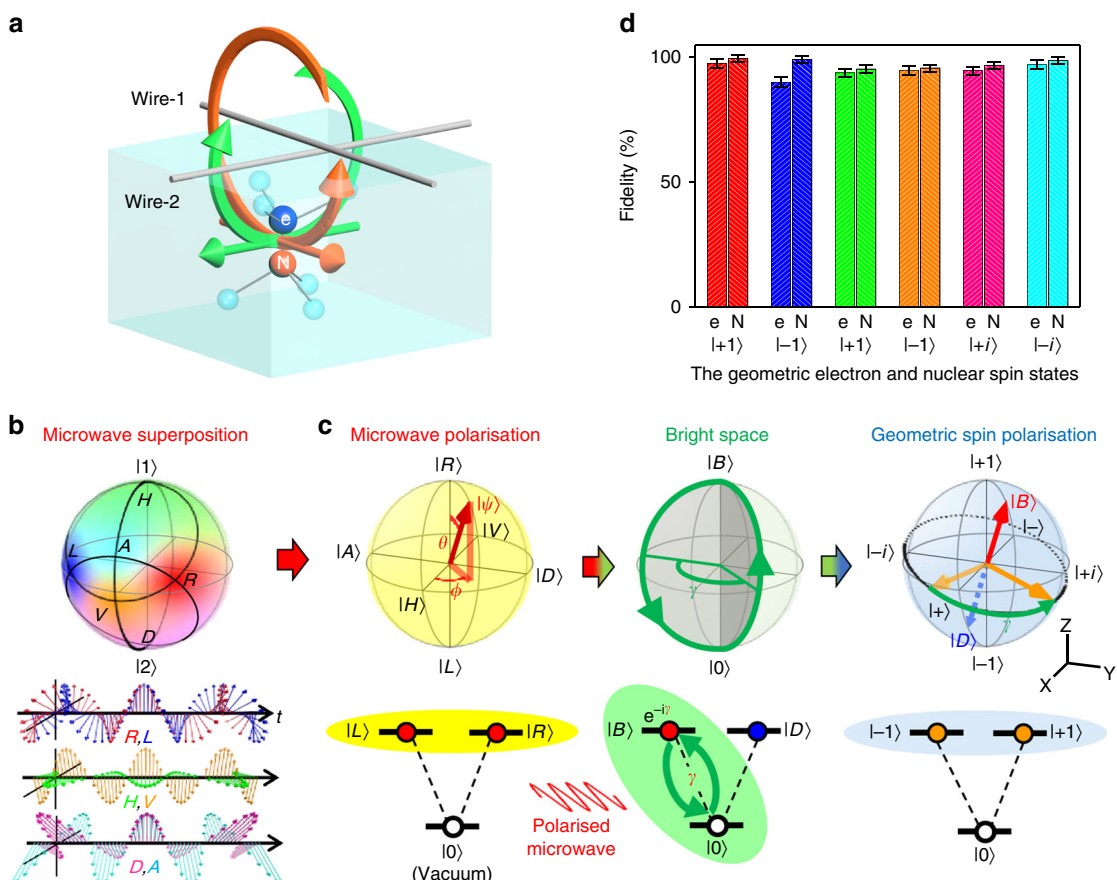

**Fig. 1** Polarisation-based holonomic quantum gates. **a** The configuration of the two crossed wires to produce a polarised microwave at the position of an NV centre. **b** Mapping from the relative amplitude and phase of the microwaves generated by the two crossed wires into the polarisations effective for the electron or nitrogen ($^{14}$N) nuclear spin in an NV centre under a realistic configuration (Methods). The $|1\rangle$ and $|2\rangle$ represent microwaves generated from, respectively, wires 1 and 2. **c** The mechanism underlying the holonomic quantum gate. The polarised microwave induces a cyclic evolution of the geometric spin in the bright space based on the bright state $|B\rangle$ and the ancillary state $|0\rangle$ with a solid angle $2\gamma$ to induce a geometric phase $\gamma$ on $|B\rangle$, resulting in the rotation of the geometric spin around an axis defined by $|B\rangle$ and the unchanged dark state $|D\rangle$. **d** The fidelities of the state preparation and measurement for the geometric electron (e) and nitrogen nuclear (N) spin. Error bars are defined as the s.d. of the photon shot noise

microwave polarisation is represented as a state vector in the geometric phase space known as the Poincaré sphere based on the degenerate $|\pm 1\rangle_p$ states corresponding to the right $|R\rangle_p$ and left $|L\rangle_p$ circular polarisations, which are two components of the spin-1 angular momentum projected along the travel direction excluding the $|0\rangle_p$ state corresponding to the vacuum. The superposition of $|R\rangle_p$ and $|L\rangle_p$ represents any polarisations, including horizontal $|H\rangle_p$ and vertical $|V\rangle_p$ polarisations. Similarly, the geometric spin polarisation of an electron (or a nucleus of nitrogen $^{14}$N) in an NV centre in diamond is also represented as a state vector in the geometric phase space known as the Bloch sphere. This sphere is based on the degenerate $m_S(m_I) = \pm 1$ (hereafter denoted $|\pm 1\rangle_{e(N)}$) states, which correspond to the degenerate $T^{\pm}$ subspace of the triplet spin states $T^+ = |\uparrow\uparrow\rangle$, $T^- = |\downarrow\downarrow\rangle$, $T^0 = \frac{1}{\sqrt{2}}(|\uparrow\downarrow\rangle + |\downarrow\uparrow\rangle)$ with spin-1 angular momentum projected along the NV axis, excluding the $|0\rangle_{e(N)}$ state corresponding to the $T^0$ state. The $|\pm 1\rangle$ states are naturally split from the $|0\rangle$ state even under a zero-magnetic field to constitute a degenerate V-type qutrit system[27] as light polarisation does. In contrast that the polarised electro-magnetic wave becomes a flying qubit with keeping the quantum interference between the $|R\rangle_p$ and $|L\rangle_p$ states, the spin-polarised electron or nucleus becomes a stationary qubit with keeping the quantum interference between the $|\pm 1\rangle$ states.

Next, we explain the mechanism underlying the holonomic gate based on the interaction Hamiltonian. Although the direct manipulation of the geometric spin is not possible because the transition between the $T^{\pm}$ states is magnetically forbidden, indirect manipulation of the geometric spin is possible with the geometric phase[1–10] acquired via a cyclic evolution in the bright space driven by a correspondingly polarised microwave (radiowave) tuned to the zero-field splitting ~2870 MHz (the nuclear quadrupole splitting ~4.945 MHz). The magnetic coupling of the polarised microwave to the spin-polarised electron is described by the interaction Hamiltonian $H = (\Omega/2)(e^{-i\phi_+}|B\rangle_e\langle 0| + e^{i\phi_+}|0\rangle_e\langle B|)$, where $\Omega$ denotes the Rabi frequency and $|B\rangle_e$ denotes the geometric electron spin in the bright state $|B\rangle_e = \cos(\theta/2)|+1\rangle_e + e^{i\phi}\sin(\theta/2)|-1\rangle_e$ corresponding to the microwave polarisation $|\psi\rangle_p = \cos(\theta/2)|R\rangle_p + e^{i\phi}\sin(\theta/2)|L\rangle_p$ (Fig. 1c). $\phi_+$ indicates the global phase defined by the phase of the driving microwave and determines rotation axis on the bright space (Methods). This indicates that the transition between the $|B\rangle_e$ and the $|0\rangle_e$ is induced while leaving the dark state $|D\rangle_e = \sin(\theta/2)|+1\rangle_e - e^{i\phi}\cos(\theta/2)|-1\rangle_e$ unchanged (Methods). The time evolution of the geometric electron spin after a cyclic transition in the bright space is represented as the holonomy matrix[3] $U_V = e^{-i\gamma}|B\rangle_e\langle B| + |D\rangle_e\langle D| = e^{-i\gamma\mathbf{n}\cdot\boldsymbol{\sigma}}$, where $\mathbf{n}$ indicates the unit gyration vector pointing to the $|B\rangle_e$ and $\boldsymbol{\sigma}$ indicates the Pauli vector. This indicates that the bright state acquires a geometric phase or holonomy $\gamma$, which corresponds to half of the solid angle enclosed by the trajectory in the bright space.

**Geometric spin state preparation and measurement**. We first calibrate the amplitude and phase of the microwaves generated by two crossed wires to precisely prepare and measure the geometric electron or nuclear spin in the basis states $|+1\rangle, |-1\rangle, |+\rangle = (1/\sqrt{2})(|+1\rangle + |-1\rangle), |-\rangle = (1/\sqrt{2})(|+1\rangle - |-1\rangle), |+i\rangle = (1/\sqrt{2})(|+1\rangle + i|-1\rangle)$, and $|-i\rangle = (1/\sqrt{2})(|+1\rangle - i|-1\rangle)$ (Fig. 1d) by the bright state preparation and projection method[28]. The achieved fidelity is $96.2 \pm 1.6\%$ on average, as evaluated by the quantum state tomography method[29,30] (Methods). Using the same method, the geometric entangled states between the electron and nitrogen nuclear spins are also prepared and measured in the

Bell states $|\Phi^{\pm}\rangle_{e,N} = (1/\sqrt{2})(|+1,+1\rangle_{e,N} \pm |-1,-1\rangle_{e,N})$ and $|\Psi^{\pm}\rangle_{e,N} = (1/\sqrt{2})(|+1,-1\rangle_{e,N} \pm |-1,+1\rangle_{e,N})$ with fidelity of $92.4 \pm 3.1\%$ on average (Methods).

**Holonomic gate process tomography**. Assuming the state preparation and measurement error is negligible, we evaluate the fidelity of the holonomic single-qubit gates over a geometric electron (nitrogen nuclear) spin. A microwave (radiowave) polarised in the $|H\rangle_p, |D\rangle_p, |R\rangle_p, (1/\sqrt{2})(|H\rangle_p + |R\rangle_p)$ is applied to drive the corresponding bright geometric spin state in the X, Y, Z, and $(1/\sqrt{2})(X + Z)$ into the $|0\rangle_{e(N)}$ state and back to the original state with a geometric phase $\pi$, which results in the holonomic X, Y, Z, and Hadamard (H) quantum gates. The $\chi$ matrices, which represent the gate transformation based on the Pauli matrices, are estimated by the quantum process tomography method[29,31] (Fig. 2a) (Methods). The fidelities of the X, Y, Z, and H gates over the geometric electron spin are respectively 99, 93, 93, and 93%, while those over the geometric nuclear spin are 97, 93, 98, and 95% (Fig. 2b).

In the same way, the holonomic two-qubit gates can also be performed by adding a geometric phase to the corresponding joint state (Fig. 3a). We demonstrate the holonomic controlled-Z (CZ) gate by driving the joint state $|-1,-1\rangle_{e,N}$ into $|0,-1\rangle_{e,N}$ and back to the original state with a circularly polarised microwave resonant to one of hyperfine-split frequencies. The fidelity degradation of the Bell states after the holonomic CZ gate is evaluated by the two-qubit quantum state tomography method (Fig. 3b). The $|\Phi^+\rangle_{e,N}$ state prepared with 98% fidelity is transformed into the $|\Phi^-\rangle_{e,N}$ state with 86% fidelity, while the $|\Psi^+\rangle_{e,N}$ state prepared with 91% fidelity remained in the same state with 90% fidelity (Fig. 3c). These results indicate that we can achieve two-qubit gate fidelity of about 90%. Any arbitrary holonomic two-qubit gates can also be constructed by combining the holonomic CZ gate with holonomic single-qubit gates. For example, the holonomic controlled-NOT (CNOT) gate can be constructed with the CZ gate and the Hadamard gate.

**Discussion**

We showed that the interaction of an arbitrary polarised microwave with a geometric electron spin qubit allows direct holonomic gate via the spin ground state as an ancillary level without the mediation of the spin−orbit interaction as in the previous optical holonomic gate[22,23] to avoid the unwanted interference and relaxation, thus enabling fast and arbitrarily precise gating at an ambient temperature. Note that polarised microwaves serve as noncommutable basis operators known as Pauli operators to satisfy the universality of the single-qubit holonomic gates[20]. This also allows the holonomic gate of a geometric nuclear spin, which was impossible by the optical gate. We also demonstrated a nontrivial two-qubit holonomic gate, which manipulates the entanglement between the electron and nuclear geometric spin, to assure the universality of the holonomic gate to perform arbitrary quantum computation tasks.

The significance of the demonstration is that the geometric spin qubit is manipulated under a completely zero field[11,22,23] to provide spatial inversion symmetry for the external field, leading to time inversion symmetry or commutability of Hamiltonians for the environmental nuclear spins[11]. The overall symmetry results in ideal manipulation of the qubit. The conventional frequency freedom should not be used since it inevitably lacks the spatial inversion symmetry by applying a magnetic field[19–21,24,25]. In contrast, polarisation freedom enables purely geometric time-reversal manipulation of the geometric spin qubit under a zero magnetic field[22,23].

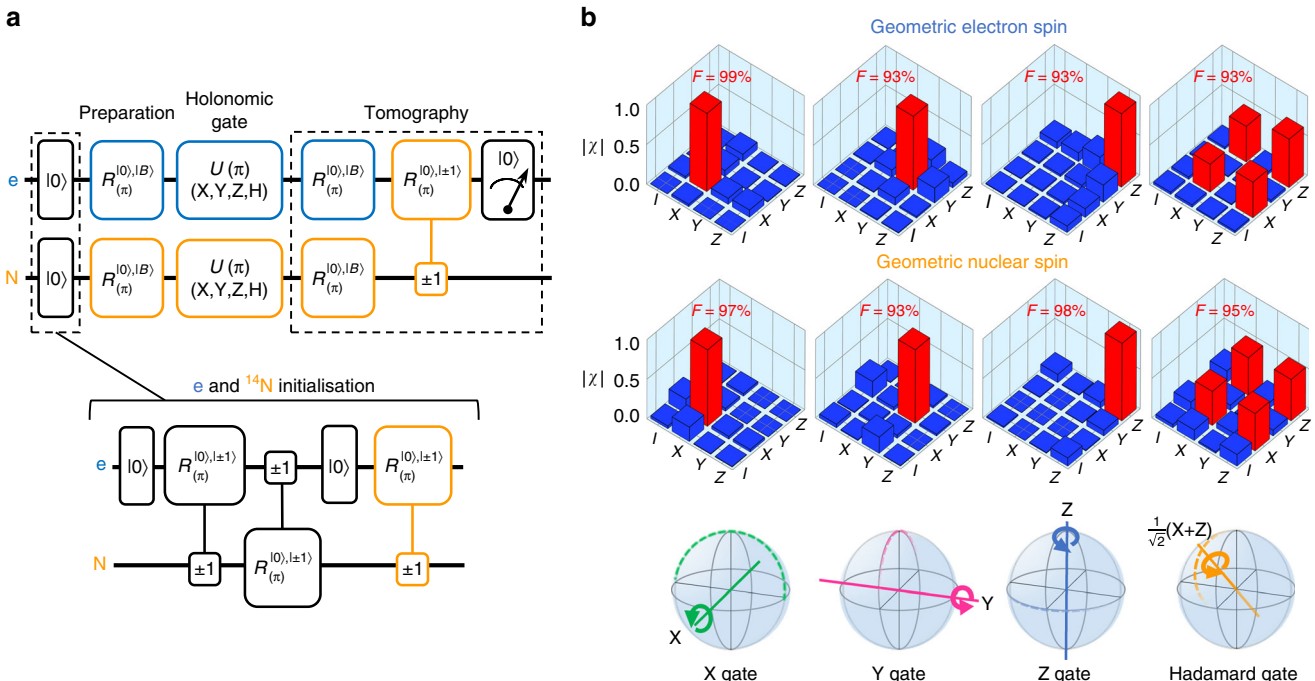

**Fig. 2** Holonomic single-qubit gates. **a** The quantum circuit of holonomic gate process tomography for geometric electron and nuclear spin qubits. The blue (orange) quantum circuits indicate the operation for the geometric electron (nuclear) spin. The gate $R_{(\pi)}^{|0\rangle,|\pm1\rangle}$ indicates the π-rotation on the target spin in the $|0\rangle - |\pm1\rangle$ space conditioned by the control spin state $|\pm1\rangle$ (double-sign corresponds) (Methods). **b** The χ matrices (absolute values) to show the fidelity of the holonomic single-qubit quantum gates for the X, Y, Z, and Hadamard (H) gates over the geometric electron and nitrogen nuclear spins reconstructed by quantum process tomography. The X, Y, Z, and H gates indicate π-rotation around the X, Y, Z, and $\frac{1}{\sqrt{2}}(X+Z)$ axes. Axes X, Y, and Z correspond to the direction defined by $|+\rangle$ and $|-\rangle$, $|+i\rangle$ and $|-i\rangle$, and $|+1\rangle$ and $|-1\rangle$, as shown in Fig. 1c. The F indicates fidelity and the bars indicate the values of the matrix elements

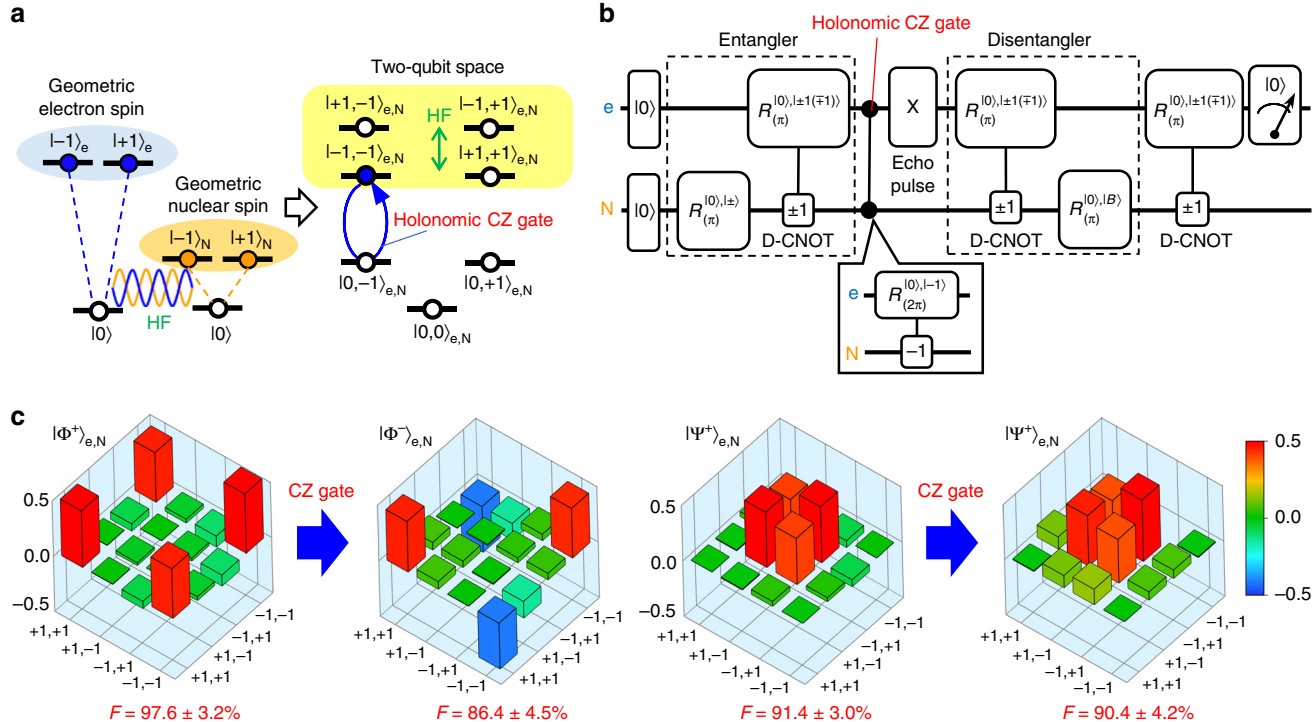

**Fig. 3** Holonomic two-qubit gate. **a** Schematic energy level diagram showing the mechanism underlying the holonomic two-qubit gate for the hyperfine-coupled electron−nuclear spins. The holonomic controlled-Z (CZ) gate is performed by flipping the phase of the $|-1, -1\rangle_{e,N}$ state with a circularly polarised microwave resonant to one of hyperfine-split frequencies. **b** The quantum circuit to evaluate the holonomic two-qubit gate by using the dynamic CNOT (D-CNOT, Methods) gate together with the geometric spin echo[11] to compensate the phase evolution during the manipulation. **c** The real part of the density matrices of the two-qubit states reconstructed by the two-qubit quantum state tomography to show the transformation by the holonomic CZ gate. The bases denoted as ±1, ±1 indicate the $|\pm1, \pm1\rangle_{e,N}$ states and the F indicates fidelity. The bars indicate the values of the matrix elements

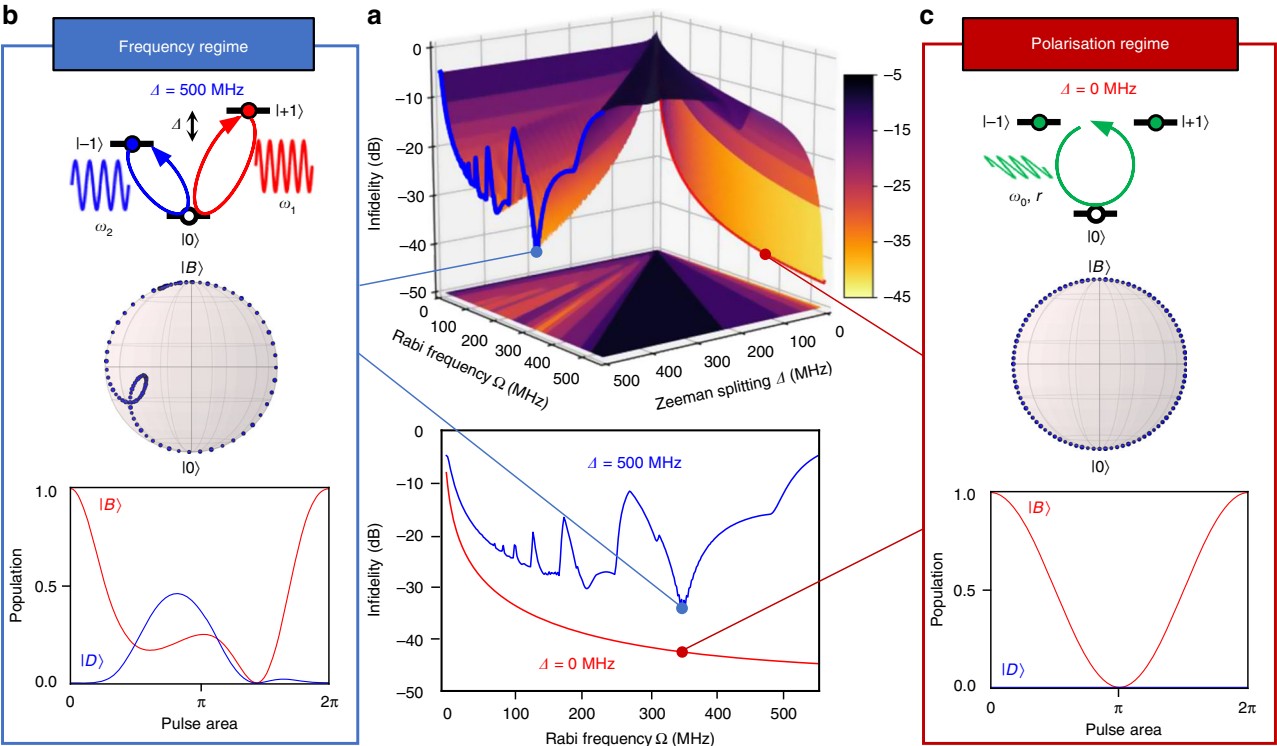

**Fig. 4** Simulated fidelities of the holonomic quantum gate. **a** The X gate infidelity (error) defined as 1−fidelity as functions of the Rabi frequency $\Omega$ and the Zeeman splitting frequency $\Delta$ of the $|\pm1\rangle_e$ states, simulated including the interference effect and dephasing using the Master equation. **b** Dynamics of the state vector in the bright space in the conventional frequency regime assuming the Zeeman splitting $\Delta = 500$ MHz and the Rabi frequency $\Omega = 350$ MHz, where the state vector does not trace a true circle due to the interference effect to involve the dark state. **c** Dynamics in the polarisation regime assuming $\Delta = 0$ and $\Omega = 350$ MHz, where the state vector traces a true circle without the interference effect to ideally exclude the dark state, showing the advantage of the polarisation regime

The degradation of gate fidelity can be attributed to dephasing due to hyperfine interaction with the environmental nuclear spins or to mutual interference of the two wires. The former can be improved by reducing the gating time while increasing the driving power. The Rabi frequency used for the geometric electron spin manipulation is about 15 MHz, that is not strong enough compared to hyperfine interaction from the $^{13}$C or $^{14}$N nuclear spins. Significant improvement in gate fidelity is expected with larger Rabi frequency[32]. This problem can be also improved by the geometric spin echo[11], and by optimising microwave pulse with the GRAPE (gradient ascent pulse engineering) algorithm[33] or composite pulses. The latter can be improved by precisely adjusting the angle so that the two wires are at a right angle to each other.

Our scheme relies only on the polarisation of a microwave instead of the frequency in the two-frequency scheme[19–21,24] to define the rotation axis. In future implementations, this difference should provide a significant advantage in gate fidelity, as shown in Fig. 4a, for the holonomic X gate on the geometric electron spin qubit under the effect of environmental nuclear spins as functions of the Zeeman splitting of the $|\pm1\rangle_e$ states and the Rabi frequency in the bright space. In the two-frequency scheme, the fidelity is expected to saturate and oscillate to make the error higher than −32 dB as the Rabi frequency increases due to the interference between the two frequencies[24], which induces a deviation of the trajectory from a true circle in the bright space to acquire the expected holonomy (Fig. 4b). Although pulse shaping technique can be used to compensate the deviation, the Rabi frequency $\Omega$ is restricted by the energy difference of two frequency $\Delta$ to satisfy $\Omega \ll \Delta$. The resulting longer manipulation time induces degradation of the fidelity by the effect of environmental nuclear spins. In

contrast, at the degeneracy point, i.e., where the Zeeman splitting diminishes and the polarisation-based holonomic gate scheme applies, the gate fidelity is expected to monotonically increase with a simple square pulse as the Rabi frequency increases to reduce the error to below −45 dB (Fig. 4c).

On the other hand, the two-qubit gate can be affected by interference even under a zero-magnetic field, since the gate relies on the hyperfine splitting. This requires a Rabi frequency smaller than the hyperfine splitting to avoid unexpected excitation, resulting in a longer gate time to deviate from the expected holonomy due to unresolvable hyperfine splitting by the environmental nuclear spins. However, the fidelity of the nuclear spin gate conditioned by the electron spin can be recovered by the geometric spin echo[11] (Methods). The two-qubit gate is also applicable to the carbon isotope ($^{13}$C) around the NV centre.

Although our scheme utilises the $|0\rangle$ state as an ancilla for the holonomic quantum gate, it can also be used to detect errors caused by the gate operation or spin relaxation to make the qubit a digital-like qubit with error detection. In the experiment, the initialisation error is excluded by heralding the $|0\rangle$ state even though the initialisation rate of the $^{14}$N nuclear spin is limited to 85%. Using this scheme, not only the controlled-phase or SWAP gates between a microwave photon and a geometric spin qubit but also the quantum state transfer from a microwave photon to a geometric spin qubit can be implemented, the same as in the case of an optical photon to a geometric spin qubit at low temperature[34]. The combination of these transfers will also enable the quantum media conversion between the superconducting qubit and a photon qubit via the geometric spin qubit.

We have demonstrated universal holonomic quantum gates consisting of single-qubit X, Y, Z, and H gates and a two-qubit CZ

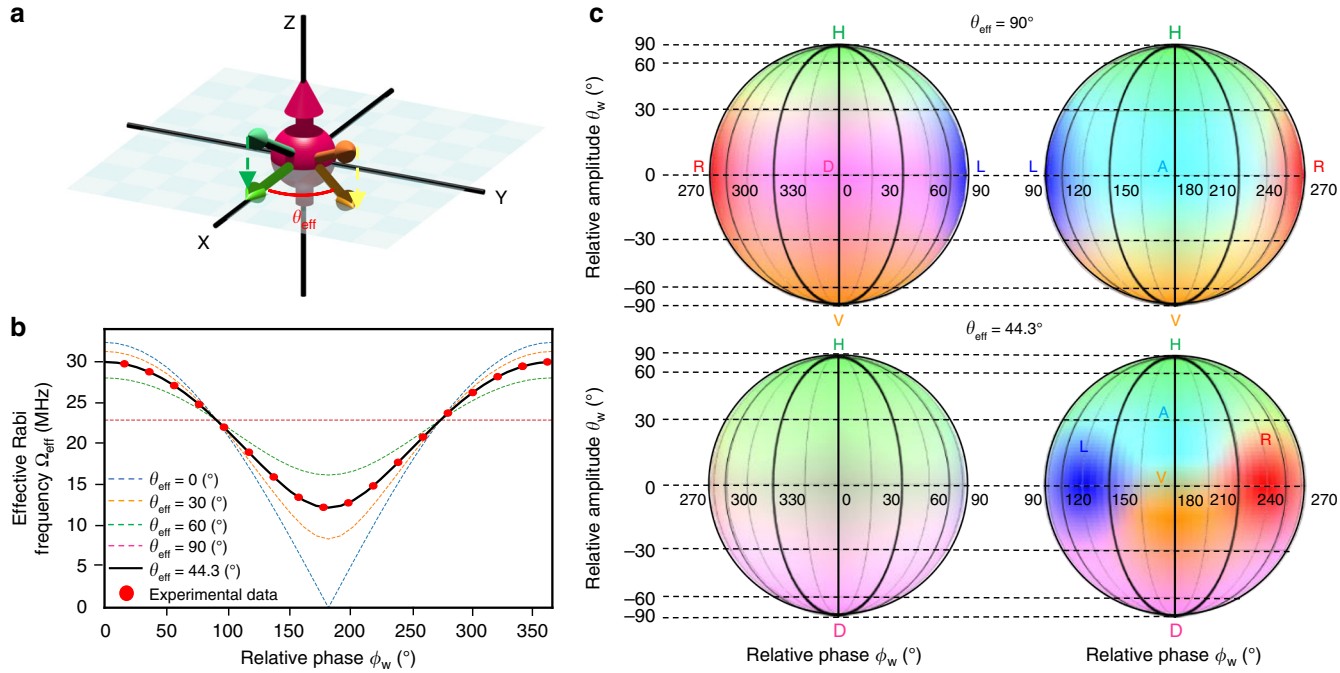

**Fig. 5** Calibration of the microwave polarisation. **a** Spatial arrangement of the field vectors of the microwaves generated from two wires and those projected in the plane normal to the NV axis (z-axis) angled by $\theta_{eff}$. The projected field vector generated by wire 1 defines the x axis. **b** The dependence of effective Rabi frequency $\Omega_{eff}$ on the relative phase $\phi_w = \phi_2 - \phi_1$ for various $\theta_{eff}$. We tuned the microwave amplitude to $\Omega_1 = \Omega_2 = 16.13$ (MHz) to see the $\phi_w$ dependence of the $\Omega_{eff}$. The experimental results indicate $\theta_{eff} = 44.3°$. **c** Mapping from the Bloch sphere spanned by the microwaves generated by the two wires parameterised by $\{\theta_w = 2\arctan((\Omega_1 - \Omega_2)/(\Omega_1 + \Omega_2)), \phi_w = \phi_2 - \phi_1\}$ to the achieved polarisations. The upper two spheres show the ideal case (effective wire angle $\theta_{eff} = 90°$), and the lower two spheres show the realistic case in our experimental case ($\theta_{eff} = 44.3°$). Even when the correspondence is largely distorted, there exists one parameter set to generate desired microwave polarisation

gate over a geometric electron and nitrogen nuclear spin qubits with polarised microwaves and radiowaves. The holonomic gates with a geometric phase other than $\pi$, such as the phase gate (S) or the ($\pi/8$) gate (T), are also achievable with this scheme. Although the H, S, T, and CZ gates offer the elementary discrete set required for the universal quantum gates to construct arbitrary unitary quantum gates, the availability of arbitrary phase gates would be beneficial for fast gating in many practical applications, such as quantum Fourier transformation and blind quantum computing[35]. The scheme allows a purely holonomic gate without requiring an energy gap, which would have induced dynamic phase interference to degrade the gate fidelity, and thus enables precise and fast control over long-lived quantum memories for realising quantum repeaters interfacing between universal quantum computers and secure communication networks.

## Methods

**Theory of geometric spin manipulation by polarised microwaves**. The Hamiltonian of an electron spin in an NV centre under the irradiation of microwaves emitted from two wires is described as follows.

$$H = D_0 S_z^2 + \gamma_e \{\mathbf{B}_1 \cos(\omega_1 t + \phi_1) + \mathbf{B}_2 \cos(\omega_2 t + \phi_2)\} \cdot \mathbf{S}, \quad (1)$$

where $D_0$ is the zero-field splitting of the electron spin, $\gamma_e$ is the gyromagnetic ratio of the electron spin, $\mathbf{B}_i$ is the magnetic field vector, $\omega_i$ is the frequency, and $\phi_i$ is the phase of the microwave generated by the wire $i = 1, 2$ and $\mathbf{S}$ is the spin-1 operator described as

$$\mathbf{S} = (S_x, S_y, S_z) = \left(\frac{1}{\sqrt{2}}\begin{pmatrix} 0 & 1 & 0 \\ 1 & 0 & 1 \\ 0 & 1 & 0 \end{pmatrix}, \frac{1}{\sqrt{2}}\begin{pmatrix} 0 & -i & 0 \\ i & 0 & -i \\ 0 & i & 0 \end{pmatrix}, \begin{pmatrix} 1 & 0 & 0 \\ 0 & 0 & 0 \\ 0 & 0 & -1 \end{pmatrix}\right). \quad (2)$$

The z-axis is defined as being in the direction of the zero-field splitting of the electron spin. Note that $\hbar$ is omitted in this formula.

We estimate that the effect of the z component of the microwave on the electron spin dynamics is 0 on average. The effective microwave fields are represented as $\mathbf{B}_1 = B_1(1, 0, 0)$ and $\mathbf{B}_2 = B_2(\cos\theta_{eff}, \sin\theta_{eff}, 0)$, where $\theta_{eff}$ is defined as the effective angle between wires 1 and 2 as shown in Fig. 5a. The Hamiltonian can then be rewritten as

$$\begin{aligned} H &\cong D_0 S_z^2 + \Omega_1 \cos(\omega_1 t + \phi_1) S_x + \Omega_2 \cos(\omega_2 t + \phi_2)\left(\cos\theta_{eff} S_x + \sin\theta_{eff} S_y\right) \\ &= D_0 S_z^2 + \left[\left\{\frac{\Omega_1 \cos(\omega_1 t + \phi_1) + \Omega_2 \cos(\omega_2 t + \phi_2) e^{-i\theta_{eff}}}{\sqrt{2}}|+1\rangle\right.\right. \\ &\quad \left.\left. + \frac{\Omega_1 \cos(\omega_1 t + \phi_1) + \Omega_2 \cos(\omega_2 t + \phi_2) e^{i\theta_{eff}}}{\sqrt{2}}|-1\rangle\right\}\langle 0| + \text{H.c.}\right], \end{aligned}$$
$$(3)$$

where $\Omega_i = \gamma_e B_i$ represents the effective intensity of the microwave with respect to the electron spin and H.c. represents the Hermitian conjugate.

By tuning the microwave frequencies of the two wires as $\omega_1 = \omega_2 = D_0$, the Hamiltonian is reduced with the interaction picture and the rotating wave approximation to

$$\begin{aligned} H &\cong \left\{\frac{\Omega_1 e^{-i\phi_1} + \Omega_2 e^{-i(\phi_2 + \theta_{eff})}}{2\sqrt{2}}|+1\rangle + \frac{\Omega_1 e^{-i\phi_1} + \Omega_2 e^{-i(\phi_2 - \theta_{eff})}}{2\sqrt{2}}|-1\rangle\right\}\langle 0| + \text{H.c.} \\ &= \left\{\frac{\Omega_+ e^{-i\phi_+}}{2\sqrt{2}}|+1\rangle + \frac{\Omega_- e^{-i\phi_-}}{2\sqrt{2}}|-1\rangle\right\}\langle 0| + \text{H.c.}, \end{aligned}$$
$$(4)$$

where $\Omega_\pm = \left|\Omega_1 e^{i\phi_1} + \Omega_2 e^{i\phi_2 \pm \theta_{eff}}\right|$ and $\phi_\pm = \arg\left(\Omega_1 e^{i\phi_1} + \Omega_2 e^{i(\phi_2 \pm \theta_{eff})}\right)$. Further calculations result in the following form:

$$\begin{aligned} H &= \frac{1}{2\sqrt{2}}\left[e^{-i\phi_+}\left\{\Omega_+|+1\rangle + \Omega_- e^{-i(\phi_- - \phi_+)}|-1\rangle\right\}\langle 0| + \text{H.c.}\right] \\ &= \frac{\Omega_{eff}}{2}\left[e^{-i\phi_+}\left(\cos\frac{\theta}{2}|+1\rangle + \sin\frac{\theta}{2}e^{i\phi}|-1\rangle\right)\langle 0| + \text{H.c.}\right], \end{aligned}$$
$$(5)$$

where $\Omega_{eff} = \sqrt{\Omega_+^2 + \Omega_-^2}$ is the effective Rabi frequency, $\phi_+$ is the global phase, $\theta = 2\arctan(\Omega_-/\Omega_+)$ is the relative amplitude, and $\phi = \phi_- - \phi_+$ is the relative phase. $\theta$ and $\phi$ correspond respectively to the polar angle and azimuth angle of the bright state vector $|B\rangle$ in the geometric spin space. The phase of the rotation axis in the bright space is specified by $\phi_+$.

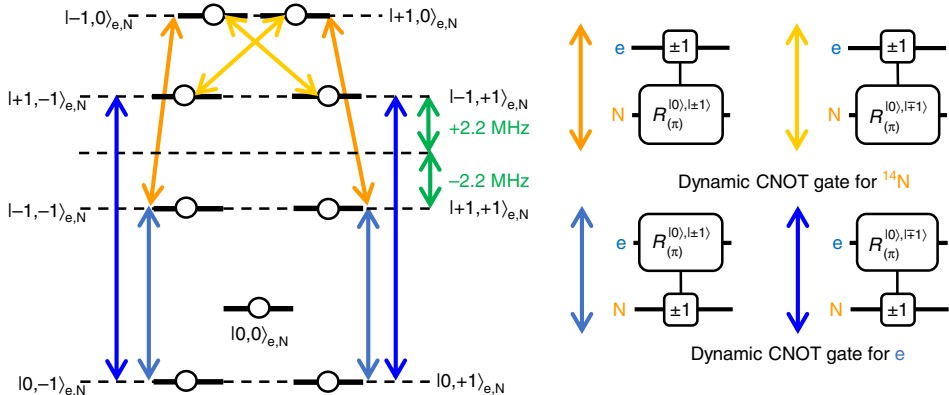

**Fig. 6** Energy-level diagram for the geometric electron and nuclear spin and the corresponding quantum circuits. The hyperfine interaction ($S_z S_z$ Ising type) induces level splitting in the $|\pm1, \pm1\rangle_{e,N}$ subspace. The even-parity states $|+1, +1\rangle_{e,N}$ and $|-1, -1\rangle_{e,N}$ form the Bell states $|\Phi^\pm\rangle_{e,N}$, and the odd-parity states $|+1, -1\rangle_{e,N}$ and $|-1, +1\rangle_{e,N}$ form the Bell states $|\Psi^\pm\rangle_{e,N}$. Note that the sign of the nuclear quadrupole splitting $Q$ of $^{14}$N is different from that of the zero-field splitting $D_0$ of an electron spin ($D_0 > 0$ and $Q < 0$)

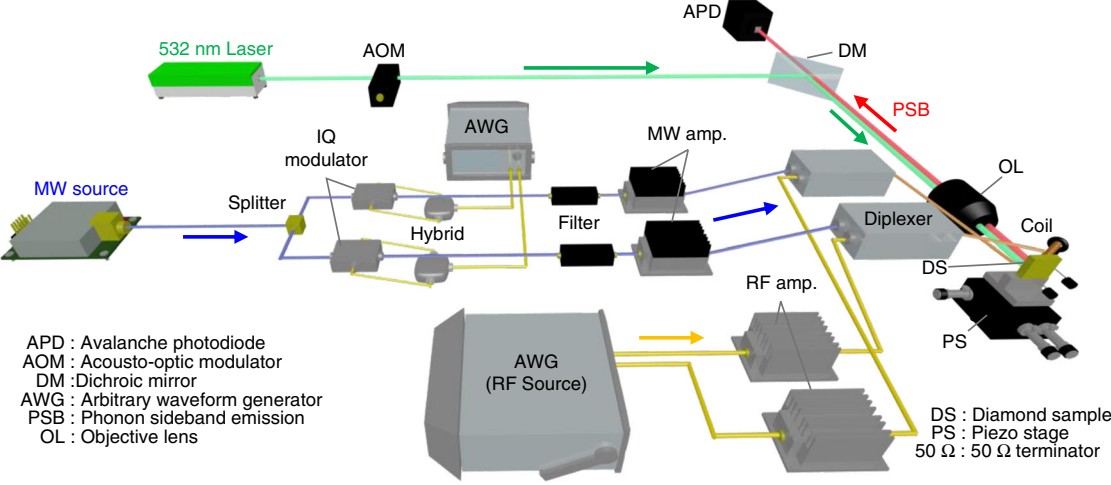

**Fig. 7** Schematic of the experimental setup

From Eq. (5), the π rotation in the bright space around the axis with phase $\phi_+$ for the state preparation and projection is represented as

$$R^{|0\rangle,|B\rangle}_{(\pi,\phi_+)} = e^{-i\left(\frac{\pi}{2}+\phi_+\right)}|B\rangle\langle0| + e^{-i\left(\frac{\pi}{2}-\phi_+\right)}|0\rangle\langle B| + |D\rangle\langle D| \qquad (6)$$

and the 2π rotation in the bright space around the axis with phase $\phi_+$ for the holonomic quantum gate is represented as

$$R^{|0\rangle,|B\rangle}_{(2\pi,\phi_+)} = e^{-i\pi}\left(|B\rangle\langle B| + |0\rangle\langle0|\right) + |D\rangle\langle D|, \qquad (7)$$

where

$$|B\rangle = \cos\left(\frac{\theta}{2}\right)|+1\rangle + e^{i\phi}\sin\left(\frac{\theta}{2}\right)|-1\rangle \qquad (8)$$

$$|D\rangle = \sin\left(\frac{\theta}{2}\right)|+1\rangle - e^{i\phi}\cos\left(\frac{\theta}{2}\right)|-1\rangle. \qquad (9)$$

The holonomic quantum gate[8–10] $U(\gamma)$ can also be achieved by sequentially applying two π rotations with the global phase $\phi_+^{(1)} = 0$ and $\phi_+^{(2)} = \gamma - \pi$ for each pulse as

$$U(\gamma) = R^{|0\rangle,|B\rangle}_{(\pi,\phi_+^{(2)})} R^{|0\rangle,|B\rangle}_{(\pi,\phi_+^{(1)})} = e^{-i\left(\pi-\phi_+^{(1)}+\phi_+^{(2)}\right)}|B\rangle\langle B| + e^{-i\left(\pi+\phi_+^{(1)}-\phi_+^{(2)}\right)}|0\rangle\langle0| + |D\rangle\langle D|$$
$$= e^{-i\gamma}|B\rangle\langle B| - e^{i\gamma}|0\rangle\langle0| + |D\rangle\langle D| \qquad (10)$$

By tuning γ, the geometric phase given for the geometric spin can be arbitrarily determined between $[-\pi, \pi]$.

The $\theta_{\text{eff}}$ can be estimated by the effective Rabi oscillation. In the experiment, the relative amplitudes of microwaves by the two wires $\theta_w = 2\arctan((\Omega_1 - \Omega_2)/(\Omega_1 + \Omega_2))$ were adjusted to be 0 ($\Omega_1 = \Omega_2 = \Omega$), and the relative phase $\phi_w = \phi_2 - \phi_1$ of the microwaves by the two wires was swept to show the phase dependence. The $\Omega_{\text{eff}}$ should become

$$\Omega_{\text{eff}} = 2\Omega\sqrt{1 + \cos\phi_w\cos\theta_{\text{eff}}}. \qquad (11)$$

The $\theta_{\text{eff}}$ in the experiment was estimated from the $\phi_w$ dependence of the $\Omega_{\text{eff}}$ to be 44.3° (Fig. 5b).

The $\theta$ and $\phi$ can be determined by controlling the $\theta_w$ and $\phi_w$ as long as the two wires are not completely parallel (Fig. 5c).

The same calculation can be applied to the $^{14}$N nuclear spin. The Hamiltonian of $^{14}$N is described as follows.

$$H_N = QS_z^2 + \gamma_N\left\{\mathbf{B}_1\cos(\omega_1 t + \phi_1) + \mathbf{B}_2\cos(\omega_2 t + \phi_2)\right\} \cdot \mathbf{S}, \qquad (12)$$

where $Q$ is the nuclear quadrupole splitting and $\gamma_N$ is the gyromagnetic ratio of the nuclear spin. Only the coefficient is different from the Hamiltonian of the electron spin.

**Dynamic CNOT between geometric electron and nuclear spin qubits.**
"Dynamic CNOT" (D-CNOT) is one of the entangling manipulations in the spin-1 system and is a conceptual extension of general (in a spin-half system) CNOT. In the presence of Ising-type interaction between the two spin-1 systems $S_z S_z$, the degeneracy is lifted by the hyperfine interaction even under a zero-magnetic field, as shown in Fig. 6. The degenerate $|+1, +1\rangle_{e,N}$ and $|-1, -1\rangle_{e,N}$ states corresponding to the even-parity Bell states $|\Phi^\pm\rangle_{e,N}$ and the degenerate $|+1, -1\rangle_{e,N}$ and $|-1, +1\rangle_{e,N}$ states corresponding to the odd-parity Bell states $|\Psi^\pm\rangle_{e,N}$ split with

each other by the hyperfine interaction, thus allowing the direct generation of the entangled state. A microwave resonant to the $|\Phi^\pm\rangle_{e,N}$ or $|\Psi^\pm\rangle_{e,N}$ states induces the transition of the electron spin (target) between the $|\pm1\rangle_e$ and $|0\rangle_e$ states depending on the nitrogen nuclear spin (control) state $|\pm1\rangle_N$. In the case of $\pi$-rotation, the D-CNOT makes direct transitions from the initial state $|0,\pm\rangle_{e,N}$ to the Bell state $|\Phi^\pm\rangle_{e,N}$ or $|\Psi^\pm\rangle_{e,N}$. The phase of the initial state is transferred to that of the entangled state. Note that all of the levels relating to the transition are degenerated

to allow the D-CNOT gate without any phase evolution unless any additional hyperfine interactions exist.

**Experimental setup.** The experimental setup is shown in Fig. 7. We used a native NV centre in a high-purity type-IIa bulk diamond grown by chemical vapour deposition and having a <100> crystal orientation (electronic grade from Element Six) without any dosing or annealing. A negatively charged NV centre located

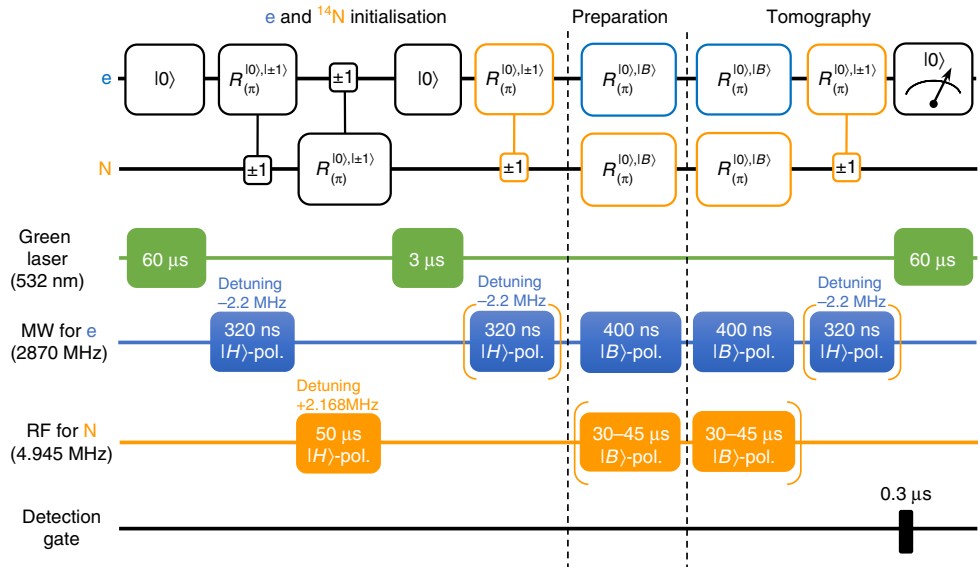

**Fig. 8** The quantum circuit and pulse sequence of single-qubit quantum state tomography for geometric electron and nuclear spin qubits. Optical initialisation of the electron spin into $|0\rangle_e$ followed by the SWAP-like gate is used for the initialisation of the $^{14}$N nuclear spin into $|0\rangle_N$. The orange quantum circuits indicate the quantum state tomography for the nuclear spin qubit, where the first D-CNOT gate is needed to eliminate imperfection in the initialisation and the second one is needed to measure the $|0\rangle_N$ state population by photoluminescence

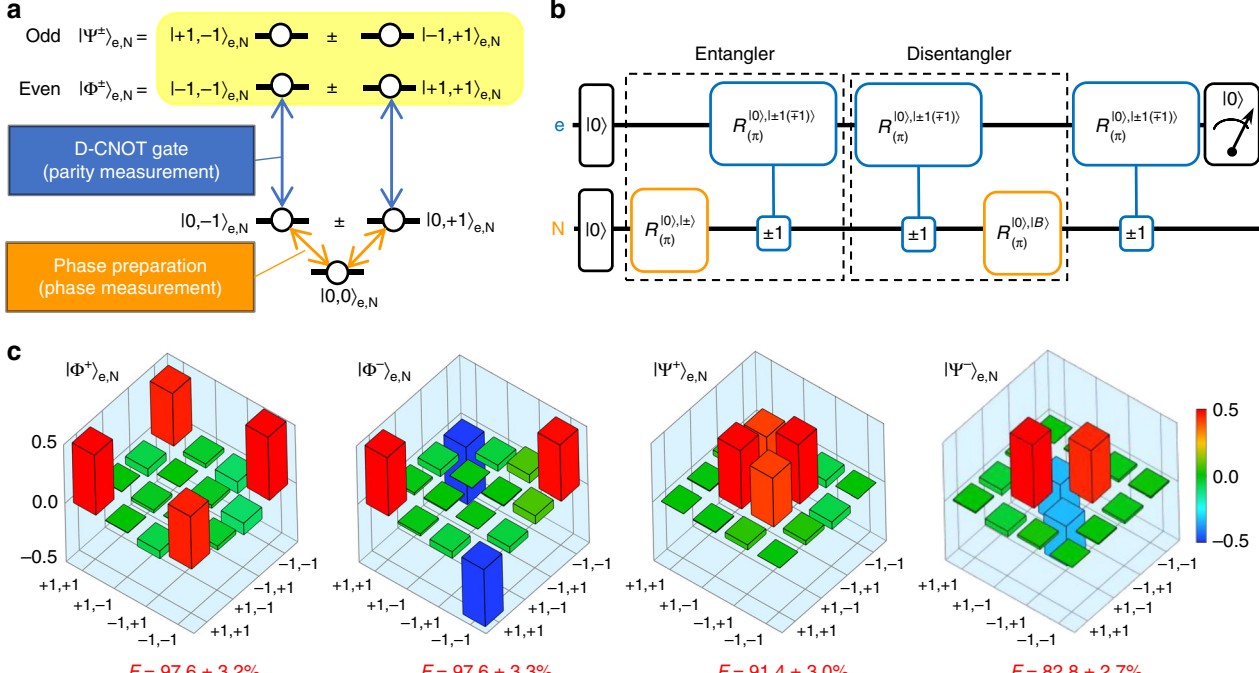

**Fig. 9** Preparation and measurement of the geometric entangled states. **a** The schematic energy-level diagram for the electron−nuclear composite system to show the scheme of the two-qubit holonomic gate. The $|\pm1\rangle_e$ and $|\pm1\rangle_N$ states are split by 4.336 MHz by the hyperfine interaction to allow frequency-based detection of parity in the entangled state whether even $|\Phi^\pm\rangle_{e,N}$ or odd $|\Psi^\pm\rangle_{e,N}$ with the D-CNOT gate, while the selection of radiowave polarisation allows phase detection. **b** The quantum circuit of the two-qubit state tomography measurement. The two $\pi$ pulses for the $^{14}$N nuclear spin correspond to the Hadamard gates used for the conventional entangler scheme. **c** The real part of the density matrix of the Bell states reconstructed by the two-qubit quantum state tomography. The F indicates fidelity and the bars indicate the values of the matrix elements

about 22 μm below the surface was found using a confocal laser microscope with a 0.8 NA ×100 objective lens. This confocal microscope was used to detect the phonon sideband (PSB) emission of the NV centre by excitation with a green laser (532 nm) for the active locking of the position and the spin state measurement. An external magnetic field was applied to carefully compensate for the geomagnetic field of about 0.045 mT using a coil magnet during the monitoring of the optically detected magnetic resonance (ODMR) spectrum. From the ODMR spectrum, the NV centre used in the experiment showed hyperfine splitting caused by a $^{14}$N nuclear spin at 2.168 MHz, and the inhomogeneous broadening of the spectrum caused by the environmental nuclear spin bath ($^{13}$C) was 0.38 MHz. All experiments were performed at room temperature.

We placed two crossed copper wires 25 μm in diameter on the surface of a bulk diamond to generate arbitrary polarised microwaves and radiowaves (Fig. 1a). The arbitrary control of the microwave properties (intensity, frequency, and phase) was possible with IQ modulation using the arbitrary waveform generator (AWG). Radiowaves were generated directly from the same type of AWG. We symmetrically duplicated these arbitrary waveform control units to perform independent and synchronised control over microwaves (radiowaves) flowing in the two crossed wires. To avoid unwanted inductive coupling between the wires, we carefully adjusted their angle to be as orthogonal as possible.

**Initialisation and readout of the spin state**. The electron spin state is first initialised into the $|0\rangle_e$ state[36] by optical pumping with a green laser for 60 μs. During the initialisation process, $^{14}$N nuclear spin is driven into the completely mixed state. The $|0\rangle_e$ state is then transferred to the $^{14}$N nuclear spin to initialise into the $|0\rangle_N$ state with two consecutive D-CNOT gates controlled by each other (Fig. 8). After that, the electron spin is initialised again by optical pumping with a green laser for 3 μs.

The photoluminescence in the PSB indicates the population of the $|0\rangle_e$ state to allow the projective measurement of the geometric spin state. We can also indirectly measure the population of the $|0\rangle_N$ state by the photoluminescence after applying the D-CNOT gate by using the joint state $|0, 0\rangle_{e,N}$.

**Preparation and measurement of the geometric spin state**. We show that the arbitrary single geometric electron or nuclear spin state is prepared and measured (projected) by the pulse sequence as shown in Fig. 8. After the electron and nuclear spin initialisation into the $|0\rangle$ state, we apply two π pulses with different polarisations. From Eq. (6), the unitary operator contributing to the state transition is written as

$$R^{|0\rangle,|B_{\mathrm{proj}}\rangle}_{(\pi,\phi_{\mathrm{proj}})} R^{|0\rangle,|B_{\mathrm{pre}}\rangle}_{(\pi,\phi_{\mathrm{pre}})} |0\rangle = \left( e^{-i(\pi-\phi_{\mathrm{proj}}+\phi_{\mathrm{pre}})}|0\rangle\langle B_{\mathrm{proj}}| + e^{-i(\frac{\pi}{2}+\phi_{\mathrm{pre}})}|D_{\mathrm{proj}}\rangle\langle D_{\mathrm{proj}}| \right)|B_{\mathrm{pre}}\rangle, \quad (13)$$

where $|B_{\mathrm{pre}}\rangle$ is the bright state of the first π pulse, $|B_{\mathrm{proj}}\rangle$ and $|D_{\mathrm{proj}}\rangle$ are the bright and dark states of the second π pulse. The first term indicates that the bright state is projected into the $|0\rangle$ state to be measured, and the second term indicates that the dark state remains dark since we can measure only the $|0\rangle$ state by photoluminescence. We can reconstruct the density matrix $\rho = |B_{\mathrm{pre}}\rangle\langle B_{\mathrm{pre}}|$ by the quantum state tomography method[29,30] with the mutually unbiased bases {$|\pm 1\rangle$, $|\pm\rangle$, $|\pm i\rangle$}. In the case of electron spin, to minimise the state preparation and measurement error (SPAM error) inherent in the state tomography, the π pulses for the preparation and projection were individually optimised by the GRAPE (gradient ascent pulse engineering) algorithm[33].

**Preparation and measurement of the geometric entangled state**. The D-CNOT gate enables the preparation and measurement of the entangled states needed for the two-qubit state tomography based on the energy levels shown in Fig. 9a and the pulse sequence shown in Fig. 9b. The real parts of the two-qubit density matrices and the fidelities compared with the expected Bell states are shown in Fig. 9c. The selection of the resonant frequency enables parity measurement, whether even $|\Phi^{\pm}\rangle_{e,N}$ or odd $|\Psi^{\pm}\rangle_{e,N}$, and the selection of radiowave polarisation enables phase measurement, whether the in-phase $|\Phi^{+}\rangle_{e,N}$ $|\Psi^{+}\rangle_{e,N}$ or the out-phase $|\Phi^{-}\rangle_{e,N}$ $|\Psi^{-}\rangle_{e,N}$. The unbalance of the fidelities between even $|\Phi^{\pm}\rangle_{e,N}$ and odd $|\Psi^{\pm}\rangle_{e,N}$ states is considered to be due to the fact that the flip-flop term of the hyperfine interaction affects only the odd states.

**Consideration of controlled-Z gate**. The CZ gate rotates the electron spin with the circularly polarised microwave controlled by the nuclear spin. The microwave needs to be weak enough not to excite the other states (Fig. 10). However, the gate fidelity becomes degraded due to the dephasing of the geometric spin qubit induced by the hyperfine interaction between the electron and environmental nuclear spins (blue line). The gate fidelity is partially recovered by the geometric spin echo[11], although the recovery is not as expected (pink line) since the dephasing effect on the targeted state is not the same as on the other state. In contrast, the gate fidelity for the nuclear spin rotation with the circularly polarised radiowave controlled by the electron spin is recovered as expected (red line) since the drive Hamiltonian is commutative with the interaction.

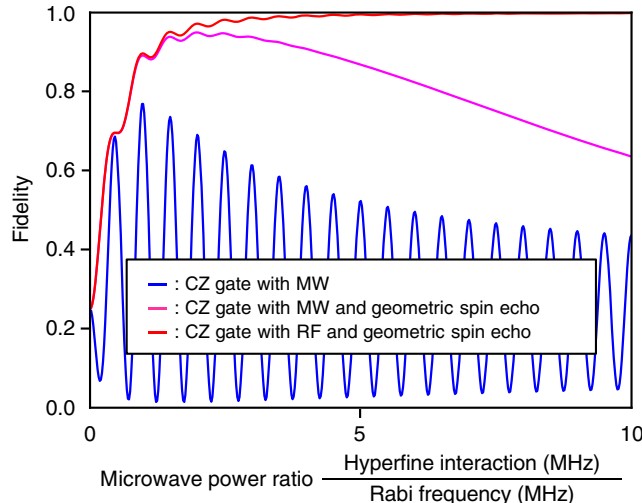

**Fig. 10** Power dependence of CZ gate fidelity. Blue, pink, and red lines represent CZ gate fidelity with, respectively, the circularly polarised microwave, with the microwave and the geometric spin echo, and with the circularly polarised radiowave and the spin echo

**Data availability**. The data that support the findings of this study are available from the corresponding author upon request.

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

## Acknowledgements

We thank Yuichiro Matsuzaki, Burkhard Scharfenberger, Kae Nemoto, William Munro, Norikazu Mizuochi, Nobuyuki Yokoshi, Fedor Jelezko and Joerg Wrachtrup for their discussions and experimental help. This work was supported by the National Institute of Information and Communications Technology (NICT) Quantum Repeater Project; by Japan Society for the Promotion of Science (JSPS) Grants-in-Aid for Scientific Research (24244044, 16H06326, 16H01052); by the Ministry of Education, Culture, Sports, Science, and Technology (MEXT) as an "Exploratory Challenge on Post-K computer" (Frontiers of Basic Science: Challenging the Limits); by the Research Foundation for Opto-Science and Technology; and by a Japan Science and Technology Agency (JST) CREST Grant Number JPMJCR1773, Japan.

## Author contributions

K.N. carried out the experiment. K.K. supported the experiment and provided theoretical framework. K.N. and K.K. analysed the data and wrote the manuscript. Y.S. provided theoretical support. H.K. supervised the project. All authors discussed the results and commented on the manuscript.

## Additional information

**Competing interests:** The authors declare no competing interests.

