## [Peer Review File · Nature Communications]

REVIEWERS' COMMENTS:

Reviewer #1 (Remarks to the Author):

The responses and revised manuscript (now submitted to Nature Communications) by Nagata and coworkers have clarified some of the points raised by the previous round of review.

As mentioned in my earlier report, the reported experiment made an interesting technical advance. In particular, it demonstrated holonomic quantum gates under both room-temperature and zero magnetic field conditions:

- The room temperature condition distinguishes this work from previous NV holonomic gate experiments using optical controls [21-23, & Opt. Lett. 43, 2380 (2018)] that require cryogenic temperatures.

- The zero magnetic field condition differentiates this work from previous experiments using Zeeman splitting [19,20].

It is still nice to see an experiment that can fulfill both conditions. In addition, the authors also extended the holonomic gates to both electron and nuclear spins. I think that these key advances might be good enough to justify publication in Nature Communications.

In the response, the authors tried to emphasize the merits of working at zero magnetic field (e.g., potentially faster gates, no interferences, etc). In principle, working at zero magnetic field provides a nice experimental design that might be important for quantum sensors, but I do not think that it is a crucial factor for implementing quantum gates. The reported data has not demonstrated obvious advantage compared to other experiments with finite magnetic field. From this aspect, I think that it is a bit weak in terms of the results obtained from this work. It would be a much stronger paper if the authors could demonstrate improved gate fidelity enabled by zero field setup as predicted in numerical simulation.

There are some statements the authors made in the response that I do not agree:

- As claimed by the authors, the key limitation is the limited MW power which can "easily reach 250 MHz with improvement microwave antenna ... ". I suspect whether it is easy to boost the MW Rabi frequency, as the control of the polarization of MW field requires new designs of the antenna and might compromise the strength of the drive.

- For non-zero magnetic field experiment, it is feasible to apply large magnetic field to induce $\Delta \sim 1$ GHz, which can enable fast MW control with Rabi ~ 250 MHz as well.

Hence, zero magnetic field is actually not a necessary feature for high-fidelity holonomic quantum gates.

There is a typo " $\Omega^2 \ll 4\Delta$ " (which should be " $\Omega \ll \Delta$ " to match the dimension), which should be corrected in both responses and revised manuscript.

Reviewer #2 (Remarks to the Author):

Report on "Universal holonomic quantum gates over geometric spin qubits with polarized microwaves", by Kodai Nagata et al., ref no NCOMMS-18-13433-T.

I have read the revised version of the paper and the authors response to the previous referee comments. The presentation has improved by clarification of relevant issues and by adding

relevant references. My concerns in my previous report (for Nature Photonics) have been addressed in an appropriate way by the authors. I believe the paper constitute an important advancement of high interest to specialists in the field of quantum computation. Therefore, I would recommend it for publication in Nature Communications.

In addition, I just came across the paper Optics Letters 43, 2380 (2018) by some of the authors of the present work. This paper looks relevant and may be cited for completeness.

NCOMMS-18-13433-T

“Universal holonomic quantum gates over geometric spin qubits with polarised microwaves”

Kodai Nagata[†], Kouyou Kuramitani[†], Yuhei Sekiguchi and Hideo Kosaka*

Reviewer #1 (Remarks to the Author):

The responses and revised manuscript (now submitted to Nature Communications) by Nagata and coworkers have clarified some of the points raised by the previous round of review.

As mentioned in my earlier report, the reported experiment made an interesting technical advance. In particular, it demonstrated holonomic quantum gates under both room-temperature and zero magnetic field conditions:

- The room temperature condition distinguishes this work from **previous NV holonomic gate experiments using optical controls [21-23, & Opt. Lett. 43, 2380 (2018)] that require cryogenic temperatures.**

- The zero magnetic field condition differentiate this work from previous experiments using Zeeman splitting [19,20]. It is still nice to see an experiment that can fulfill both conditions. In addition, the authors also extended the holonomic gates to both electron and nuclear spins. I think that these key advances might be good enough to justify publication in Nature Communications.

In the response, the authors tried to emphasize the merits of working at zero magnetic field (e.g., potentially faster gates, no interferences, etc). In principle, working at zero magnetic field provides a nice experimental design that might be important for quantum sensors, but I do not think that it is a crucial factor for implementing quantum gates. The reported data has not demonstrated obvious advantage compared to other experiments with finite magnetic field. From this aspect, I think that it is a bit weak in terms of the results obtained from this work. It would be a much stronger paper if the authors could demonstrated improved gate fidelity enabled by zero field setup as predicted in numerical simulation.

There are some statements the authors made in the response that I do not agree:

- As claimed by the authors, the key limitation is the limited MW power which can "easily reach 250 MHz with improvement microwave antenna ... ". I suspect whether it is easy to boost the MW Rabi frequency, as the control of the polarization of MW field requires new designs of the antenna and might compromise the strength of the drive.

- For non-zero magnetic field experiment, it is feasible to apply large magnetic field to induce $\Delta \sim 1$ GHz, which can enable fast MW control with Rabi ~ 250 MHz as well. Hence, zero magnetic field is actually not a necessary feature for high-fidelity holonomic quantum gates.

There is a typo " $\Omega^2 \ll 4\Delta$ " (which should be " $\Omega \ll \Delta$ " to match the dimension), which should be corrected in both responses and revised manuscript.

Response for Reviewer #1

We greatly appreciate referee's kind comments and suggestions for clarifying the advantage of the demonstrated scheme against previous ones. According to his/her suggestion, we cited the following paper in reference;

Ref. 23: Ishida, N. *et al.* Universal holonomic single quantum gates over a geometric spin with phase-modulated polarized light. *Opt. Lett.*, **43**, 2380-2383 (2018).

We also corrected the typo in page 11, line 180 in the revised manuscript:

Reviewer #2 (Remarks to the Author):

Report on "Universal holonomic quantum gates over geometric spin qubits with polarized microwaves", by Kodai Nagata et al., ref no NCOMMS-18-13433-T.

I have read the revised version of the paper and the authors response to the previous referee comments. The presentation has improved by clarification of relevant issues and by adding relevant references. My concerns in my previous report (for Nature Photonics) have been addressed in an appropriate way by the authors. I believe the paper constitute an important advancement of high interest to specialists in the field of quantum computation. Therefore, I would recommend it for publication in Nature Communications.

In addition, I just came across the paper Optics Letters 43, 2380 (2018) by some of the authors of the present work. This paper looks relevant and may be cited for completeness.

Response for Reviewer #2:

I greatly appreciate referee's kind comments and suggestions. According to his/her suggestion, we cited the following paper in reference;

Ref. 23: Ishida, N. *et al.* Universal holonomic single quantum gates over a geometric spin with phase-modulated polarized light. *Opt. Lett.*, **43**, 2380-2383 (2018).